# Pregnancy by Oocyte Donation: Reviewing Fetal–Maternal Risks and Complications

**DOI:** 10.3390/ijms241813945

**Published:** 2023-09-11

**Authors:** Erica Silvestris, Easter Anna Petracca, Michele Mongelli, Francesca Arezzo, Vera Loizzi, Maria Gaetani, Pierpaolo Nicolì, Gianluca Raffaello Damiani, Gennaro Cormio

**Affiliations:** 1Gynecologic Oncology Unit, IRCCS Istituto Tumori “Giovanni Paolo II”, 70124 Bari, Italy; easteranna97@hotmail.it (E.A.P.); vera.loizzi@uniba.it (V.L.); gennaro.cormio@uniba.it (G.C.); 2Obstetrics and Gynecology Unit, University of Bari “Aldo Moro”, 70121 Bari, Italy; michelemongelli1992@gmail.com (M.M.); mariagaetani.83@libero.it (M.G.); pierpaolonicoli@virgilio.it (P.N.); damiani14@alice.it (G.R.D.); 3Department of Interdisciplinary Medicine (DIM), University of Bari “Aldo Moro”, 70121 Bari, Italy; francesca.arezzo@uniba.it; 4Department of Precision and Regenerative Medicine—Ionian Area, University of Bari “Aldo Moro”, 70121 Bari, Italy

**Keywords:** advanced techniques, assisted reproductive technology, infertility, oocyte donation, reproductive diseases

## Abstract

Oocyte donation (OD) has greatly improved over the last three decades, becoming a preferred practice of assisted reproductive technology (ART) for infertile women wishing for motherhood. Through OD, indeed, it has become possible to overcome the physiological limitation due to the ovarian reserve (OR) exhaustion as well as the poor gamete reliability which parallels the increasing age of women. However, despite the great scientific contribution related to the success of OD in the field of infertility, this practice seems to be associated with a higher rate of major risky events during pregnancy as recurrent miscarriage, infections and placental diseases including gestational hypertension, pre-eclampsia and post-partum hemorrhage, as well as several maternal–fetal complications due to gametes manipulation and immune system interaction. Here, we will revisit this questioned topic since a number of studies in the medical literature focus on the successful aspects of the OD procedure in terms of pregnancy rate without, however, neglecting the risks and complications potentially linked to external manipulation or heterologous implantation.

## 1. Introduction

Infertility is a prevalent problem in our society today. According to the World Health Organization (WHO), in the recent few decades, the number of infertile couples has greatly increased to 48 million and the number of individuals, both females and males, affected by infertility has improved to 186 million [1]. In this context, during the past 20 years, Assisted Reproductive Technologies (ART) have become a globally widespread treatment modality for infertile couples, and adoption of modern infertility procedures including in vitro fertilization (IVF) has been progressively reduced to approximately one-tenth the number of infertile women with premature ovarian failure (POF) [2].

In reality, 40 years after its disputed clinical introduction, IVF is available as a successful treatment for infertility almost all over the world. At least 2.5 million treatments are performed every year worldwide resulting in more than 500,000 births annually, while until today, over 8 million children were globally born by IVF [3]. Such a preferred procedure is still considered the most successful ART practice compared to other techniques such as intrauterine insemination and includes diversified biological and medical procedures for healthy egg fertilization while avoiding major inheritable genetic disorders [4]. In fact, IVF, as the best-registered procedure in reproductive medicine, includes several steps starting from hormonal stimulation with exogenous gonadotropins, recruitment of own or donors’ eggs for autologous or heterologous IVF, respectively, insemination, fertilization and embryo culture, until transferring embryos into the woman’s uterus [4].

However, a number of conditions are directly related to better results. In ART, as well as in natural pregnancies, the woman’s age still remains a pivotal factor in determining pregnancy success and the 80% success rate of IVF has been generally reported in females within the fourth decade of life [5]. Nevertheless, despite satisfactory results even in promoting pregnancies in older women, this procedure is not completely safe and risk-free in particular for embryos, in relation to the physical stress due to both manipulation of gametes and culture conditions, capable of generating both genetic and epigenetic derangements during the full procedure. This is a debated topic today, in particular during the current precision medicine era, which the scientific community of both gynecologists and IVF-related physicians urgently claiming to provide satisfactory answers to women undergoing ART.

Here, we provide an attempt to revisit safety conditions and risks in gametes’ employment for ART, based on the regulatory norms and effectiveness of IVF programs in providing healthy pregnancies and newborns as reported in the medical literature.

## 2. Egg Donation in Europe and Worldwide

The most recent annual reports of the European IVF-Monitoring (EIM) consortium, in agreement with the European Society of Human Reproduction and Embryology (ESHRE), have recently collected data from 39 European countries describing treatments employing donor gametes (DG), which have reached up to 8% of all ART cycles [6].

In the last decade, Spain confirmed its role as the first European country performing ART with DG with 37,618 egg donation (ED) cycles yearly, followed by Russia with 9804 cycles [6]. In the last four years, ED treatments in Italy have definitely increased since from being one of the ultimate countries using this technique in ART, Italy become the third European state per ED cycles number, after Russia, as reported by ESHRE. In fact, data from this scientific society show that in 2014 Italy accounted only for the 0.2% of the total European ED cycles, with 156 performances [7], whereas in 2018 this number raised to 5947 cycles, namely 7.3% of the total ED treatment in Europe [6].

In 2018, European ED number estimation included 36,938 donations and 24,148 of them were employed for fresh embryo transfer (ET), whereas 16,130 procedures included frozen oocytes. However, these data are underestimated since several country performances are missing. Overall 25,760 deliveries were reported with ED. Furthermore, pregnancy rate per ET is higher by freshly donated (49.6%) than for thawed oocytes (44.9%), thus preferring fresh oocytes as the best choice when using this technique [6]. However, by comparing these rates over the years, it can be appreciated that frozen eggs’ employment has grown from 37% to 44%, whereas fresh eggs’ employment is constantly around 50%.

IVF treatment success with patient’s egg employment decreases progressively with the woman’s age in relation not only for the low number but also for the reduced availability of healthy oocytes. Since just 25 countries set age limitations for egg donors generally between 18 and 35 years [8], ED is provided from young and fertile women to ensure better outcomes for their good quality, as reported by the ART Fertility Clinic and National Summary Report, leading to similar outcomes independently from recipient’s age [9]. ESHRE data about PR and delivery rate (DR) in <34, 35–39 and ≥40 yrs recipient women treated with ED, range, respectively, from 43.2% to 44.9% and from 29.5% to 33.4% [6]. Nevertheless, the risk associated with late pregnancies should not be underestimated and is a necessary additional concern not only for ED, but in all ART treatments.

Moreover, deliveries are also increased between 2014 and 2018 and this is primarily due to the higher contribution of Spain with respect to other countries [6,7]. On the other hand, IVF outcomes appear characterized by high variability, also considering that data collection can be either voluntary or compulsory, and that in some countries such as Bosnia and Herzegovina, Germany, Norway, Switzerland and Turkey, ED cannot be adopted for legal reasons. Furthermore, in Croatia, Ireland, Montenegro and Italy, no local donations are accepted although the practice is legally permitted [8]. Thus, the next objective in the ART field will focus on collecting complete data sets as well as expanding the criteria for accessibility, and increasing financial support to couples within the European countries.

## 3. Major Risky Events for Fetus in IVF-Assisted Pregnancy

### 3.1. IVF and Recurrent Miscarriages

Recurrent miscarriages (RMs) are a critical issue in public health, often responsible of psychological troubles in women’s health and inevitably within the couple relationship. Beyond chromosomal aneuploidy [10,11], several immunological [12,13] and thrombophilic [14,15] maternal factors, namely expanded population of blood natural killer (NK) cells, antiphospholipid and antithyroid antibodies, altered levels of cytokines, and more, have been associated with RMs. Since mechanisms leading to recurrent gestational loss, are still unclear, in last decade, several couples preferred to consult medically assisted procreation centers to achieve pregnancy thus increasing the rate of ART procedures.

However, despite the IVF programs that allow one to perform good-quality ET by pre-implantation genetic testing for aneuploidy (PGT-A), many infertile couples undergo recurrent implantation failure with RMs and fail in having pregnancies [16]. A systematic review and meta-analysis [17] have recently reported that an increase in maternal age is strictly associated with a decline in ART success rates independently from embryo ploidy, as well as in infertile women affected by chronic endometritis and particularly in those with recurrent implantation failure at IVF [18,19]. In a prospective case–control study, Bilibio and co-workers investigated the main reasons of RMs after spontaneous gestation as compared to IVF procedure [20] and identified several putative factors associated with RMs in 87.9%, with respect to 13.8% for which an apparent cause is lacking. However, the identified factors appeared similar in RMs after spontaneous gestation and IVF, that were then classified as immune (13.8%), thrombophilic (20.7%), both autoimmune and thrombophilic (24.1%) and genetic (29.3%) factors. Thanks to PGT-A and transfer of euploid embryos [21,22], IVF reduces to approximately one-third (29.3%) the abortion events among patients with RMs, thus generally improving IVF pregnancy outcomes. Therefore, couples that undergo IVF with a repetitive loss history should be investigated for presence of other factors than the aforementioned factors associated with RMs.

Intravenous immunoglobulin (IVIg) is an established treatment for many autoimmune and inflammatory diseases [23]. Despite inconsistent conclusions derived from meta-analyses [24,25] exploiting IVIg in RM patients, it seems to represent a chance for couples that failed to achieve a successful pregnancy after IVF and/or in “unexplained” infertility cases, often related to immunological problems, as reported by Virro [26]. However, further studies will be necessary before considering IVIg clinical employment. On the other hand, glucocorticoids are currently used in patients with autoimmune and inflammatory diseases to modulate the immune system activity and their role in IVF/ICSI procedure has been tested in a few controlled though heterogenous trials, which were combined in a meta-analysis [27]: the Authors found a borderline, though statistically significant increase in pregnancy rates after peri-implantation by glucocorticoid administration in IVF, but not in ICSI patients. Finally, immunomodulation through IVIg in combination with prednisone seems to be a promising approach to prevent recurrent pregnancy loss after ART [28].

### 3.2. IVF and Fetal–Maternal Risks

Conception is not the only concern for infertile couples with a desire of parenthood. Although, a successful ART cycle is not flawless and is associated with an increased risk of adverse maternal and perinatal outcomes as compared to spontaneous gestation, the procedure has been reported as associated with increased risk of gestational hypertension, pre-eclampsia, diabetes mellitus, intrahepatic cholestasis, placenta previa, placental abruption, preterm rupture of membranes, placental adherence, postpartum hemorrhage, polyhydramnios, preterm labor, low birth weight, and small-for-date infant [29].

It is conceivable that a large part of pregnancy complications associated with fertility treatment is due to multiple gestations that are more frequent in ART conceptions. However, the high risk of placental complications after fertility treatment appears to be independent of multiple gestations [30]. IVF has been demonstrated to be strongly associated with preterm ischemic placental disease [31], which represents a group of pathological conditions including pre-eclampsia, abruption and intrauterine growth restriction, with a still unclear pathophysiology, but likely related to inadequate remodeling of spiral arteries in early pregnancy [32]. The stronger association with preterm ischemic placental disease suggests an association between IVF and placental insufficiency [31].

Although some meta-analyses have shown that ART is associated with an increased risk of pre-eclampsia, as reported in Figure 1 [33], the underlying mechanism is still not well understood [34,35,36].

The types of ART protocols also appear to play a role in differences in maternal and perinatal outcomes [37]. In fact, a recent meta-analysis by Rogue and co-workers showed that frozen/thawed ET are related with a higher rate of low birth weight and pre-eclampsia when compared to fresh ET [38]. A recent meta-analysis by Hui Ju Chih confirmed that chances of hypertensive disorders in pregnancy and pre-eclampsia were higher in pregnancies by frozen ET and OD, also showing that IVF/ICSI probabilities of hypertension and pre-eclampsia were higher than in spontaneous conceptions irrespective of the plurality [39].

Gestational diabetes has also been studied as an outcome in IVF pregnancies because it predisposes to a higher risk of pregnancy-related hypertension, fetal macrosomia, operative delivery, and cesarean delivery [40]. An increased incidence of diabetes in pregnancy conceived via ART has been demonstrated [28,41], suggesting that ART procedures can contribute to induce alterations at molecular and/or cellular levels that promote the development of gestational diabetes.

Also, an increased risk of preterm delivery appears to recur in pregnancies conceived through IVF [42]. Particularly, the ART procedure may contribute to the risk of preterm delivery in singleton gestations: indeed, there was no difference in rate of preterm birth between twin pregnancies conceived with IVF/ICSI or without medical assistance, which is likely due to the higher risk of preterm delivery in twin pregnancies in general, making it difficult to identify a real difference, if any [43]. However, the risk of preterm delivery appears to be lower in the setting of frozen ET [43].

Abnormalities in birth weight, namely as low as less than 2500 g, have been associated with IVF [40]. The association between IVF/ICSI conception and low birth weight can be dependent upon the supraphysiological hormonal environment of the IVF cycle. A study by Kalra and coworkers [44] compared the birth weight of singleton infants born after IVF with fresh ET versus singletons born after IVF with frozen/thawed ET resulting in higher risk of low birth weight in the first group. However, maternal subfertility provides an independent risk for low birth weight in newborns [45].

Several studies highlighted a concern for potentially increased risks of primary defects in ART pregnancies. Hansen and coworkers in their meta-analysis showed a moderate increase in the risk of birth defects in ART-conceived children compared with naturally conceived ones [46]. Another study also demonstrated that the association between IVF/ICSI pregnancies and congenital malformations for singleton births is apparently stronger than that for multiple births [47]. Anyway, in accordance with previous studies, a recent work of Zhu and colleagues [48] shows that, considering IVF and ICSI as different ART subgroups, there are no risks differences for birth defects between children born with one or the other technique. Other evaluations of the correlation between birth defects and ART confirm a specific increase in the rate of congenital heart defects (CHDs) [49,50]. It is estimated that the prevalence of CHDs in children born through natural conception is 6.8 per 1000 births, while the rate in children born after ART is almost double and accounts for 13 per 1000 births [51].

Other interesting topics are imprinting disorders and neurodevelopmental disorders in IVF pregnancies, but data are still lacking and questionable [52,53].

Finally, there is evidence that at least some IVF-conceived children may be at increased risk for cardiometabolic disorders including insulin resistance, higher blood pressure and higher body fat percentage compared to children conceived without medical assistance. Most IVF children are still in their first three decades of life and, thus, limited data exist regarding long-term morbidity and mortality [54].

## 4. Recurrent Complications in IVF Pregnancies by Oocyte Donation

### 4.1. Alterations due to Manipulation of Gametes including Epigenetic Derangements

Epigenetic derangements are crucial in gametogenesis. In particular, DNA methylation is involved in different stages of oocyte maturation and seems to be strongly influenced by hormonal stimulations mainly required for multiple follicular growth induction related to ART. Thus, ART procedures are apparently associated with increased imprinting and DNA methylation errors [55]. Exploring epigenetic alterations is therefore essential to understand the occurrence of hereditary transmission of possible clinical disorders [55].

The DNA of gametes, is sensitive to the effects of oxidative stress through methylation of nucleotide bases, specifically in the CpG islands. Cytosine and guanine suffer the higher effects, with the latter being the most susceptible. The products of their oxidation are 8-oxo deoxyguanosine and 5-hydroxyme-thylcytosine [56]. Moreover, thiamine formation, which derives from their deamination, inhibits the binding of methyl CpG-binding domain proteins (MBDs) to CpG islands. This process can lead to the deranged functionality of transcription factors [57], thus resulting in heritable epigenetic changes through chromatin rearrangements [58,59]. By adopting modern molecular biology techniques to explore the gametes’ viability and health, it is possible to assess in oocytes the methylation integrity by measuring the expression of DNA methyltransferases (DNMT) expression, a major transcript responsible for DNA methylation maintenance. In fact, the methylation integrity can be assessed by measuring both DNMT-RNA as well as methyl-CpG binding proteins’ RNA levels as a linear expression of epigenetic reliability to exclude putative derangements due to physical and culture handling of gametes [57,58,59].

Furthermore, controlled ovarian stimulation required in ED cycles also increases the homocysteine levels with an associated inhibition of oxidative stress and methylation in follicular fluid, resulting in an inducing in epigenetic derangements in the zygote [60,61]. To support the importance of the epigenetic derangements in gametes’ manipulation, several authors hypothesize that the gene methylation alterations could even generate neurologic diseases, including autism, in newborns conceived through ART protocols [62].

### 4.2. Abnormal Embryo Implantation

Pregnancies from ED generate an immunological paradox since the fetus is allogeneic to the mother and hence exposes non-self antigens and cells [63]. Nevertheless, a complex combination of hormones, cytokines and immune, as well as non-immune cells, allows immunological fetal tolerance. Compared with spontaneously conceived pregnancies, there is a higher level of antigenic dissimilarity in ED cases. In fact, considering the five most immunogenic HLA antigens (HLA-A, -B, -C, -DR, and -DQ), the maximum number of mismatches in spontaneous pregnancies would be five, while being up to ten for heterologous pregnancies, resulting therefore in a greater maternal immune response. Increased levels of intracellular IFN-γ, TNF-α and IL-4-positive CD4^+^ T lymphocytes were also demonstrated as compared to spontaneous or IVF pregnancies, thus indicating a different stimulation of intracellular cytokines [64]. Moreover, the endometrial excess of NK during implantation has been demonstrated to contrast the embryo setting in uterus by an apoptotic mechanism based on the presentation of death signals to blastomeres [65].

### 4.3. Latent Endometritis

Alterations in the immunological pattern also impacts on placenta morphology in OD pregnancies. Notably, villitis of unknown etiology and chronic deciduitis have both been found to be associated with chronic inflammation [66]. From a pathologist’s point of view, the idiopathic villitis is related to the excess of chronic inflammatory cells, as cytotoxic lymphocytes and macrophages, which infiltrate the stroma of terminal villi in absence of histological evidence of an infective cause [67,68]. The chronic deciduitis is diagnosed when a chronic infiltrate by the same inflammatory cells detected in the basal decidua [66,68]. Furthermore, approximately 90% of placentas from OD pregnancies were found to contain syncytial knots representing the last stage of the apoptotic cascade in the syncytiotrophoblast. These are protrusions of the apical membrane of the syncytiotrophoblast that contain old and late apoptotic nuclei. Once the apical cytoplasmic membrane of the syncytiotrophoblast is released, the syncytial nodes are pulled into maternal circulation [69]. Apoptosis of the villous trophoblast, resulting in the formation of syncytial nodes, is upregulated in pre-eclampsia [70,71].

### 4.4. Role of Abnormal Placentation

In women with pre-eclampsia in naturally conceived pregnancies, the thrombomodulin serum levels degradation products are higher than in uncomplicated naturally conceived pregnancies [72]. Thrombomodulin is an essential protein for endothelium stability as it inhibits inflammatory pathways, endothelial cell apoptosis, and most importantly inhibits coagulation [73]. Placental thrombomodulin-like pathways are currently not completely known, but angiogenic derangement, as in pre-eclampsia, has been shown to reduce thrombomodulin expression (Figure 2) [74]. Bos and coworkers demonstrated that placental thrombomodulin expression is low in both uncomplicated OD pregnancies and in those complicated by the occurrence of pre-eclampsia, which might contribute to an increased likelihood of developing this disease after OD [75].

## 5. Immune System Interaction and Deregulation

Even if pregnancy is a physiological condition, trophoblasts and decidua play an important role in maternal–fetal tolerance, in order to avoid possible risk related to any immunological rejection. After ranging decidua, trophoblasts differentiate into villi (without HLA molecules on their surface) and extravilli (EVT) which express polymorphic HLA-C both of maternal and paternal origin, and non-polymorphic HLA-E and HLA-G, responsible for non-specific immune response.

HLA-C molecules act as ligand for killer immunoglobulin-like receptors (KIRs) on decidual NK cells and the different combination between KIR aplotypes and HLA-C allotypes is responsible for immune tolerance against fetus or increased risk of pregnancy complication [76].

Furthermore, basal level of immunoactivation is necessary, since decidual NK cells take part to spiral artery remodeling and produce angiogenic factors necessary for placentation [76,77,78]. HLA-E and HLA-G permit EVT’s direct interaction with both dNK and CD8^+^ cells, suppressing their cytotoxic activity and establishing allorecognition [76]. Decidual antigen-presenting cells (APCs), instead, express paternal HLA-C peptide fragments to CD4^+^ cells through Major Histocompatibility Complex-II (MHC-II) molecules: this interaction represents the so-called indirect antigen recognition between mother and fetus (Figure 3). It seems that the CD4^+^-activated cells number increases as HLA-C mismatch occurs [76,79].

In the maternal–fetal tolerance process, T-regulatory-cell (Treg) response is crucial for successful implantation [80] and to prevent pregnancy-associated risk. In particular, Treg participation consists of the priming of seminal plasma, non-inherited maternal antigen (NIMA)-specific Treg cell functioning, and pre-existing autologous antigen-specific Treg cells involvement [76].

In contrast with natural pregnancies, ED implies the mother’s immune system overexposure to several non-self HLA-C antigens (paternal and donor’s ones) that seem to induce higher PE incidence, but also a decrease in live birth rate, especially after double ET [81,82]. Uncomplicated OD pregnancies as well as complicated ones show a placental expression reduction of several mRNA molecules such as CD45, CD55, and CD59, which act as complement regulatory proteins [83]. Saito and colleagues [76] reported that a degradation product of complement factor C4, called C4d, could accumulate if antigen-mediated allograft rejection occurs. C4d can also be found in women with pre-eclampsia in both spontaneous and complicated OD pregnancies, but not in non-complicated ones, probably suggesting its specific association to the disease. Furthermore, pre-eclampsia in natural pregnancies seems to be associated with a decreased expression of CD68^+^ macrophages, CD4^+^ T cells and Treg cells, while in OD ones, either complicated or uncomplicated, this condition occurs independently from the occurrence of pre-eclampsia. Moreover, inflammatory lesions and maternal M2 macrophages found in the chorionic plate were linked to 0% of pre-eclampsia incidence in OD pregnancies, while a high incidence of pre-eclampsia is apparently associated with the absence of inflammatory signs, thus hypothesizing that the potential inflammatory state could perhaps avoid the fetus rejection [84].

Since the reliable immune system function is related to good pregnancy outcomes even in the case of donor gametes, it is interesting to investigate how it could react to an embryo graft in pregnant women affected by autoimmune diseases (ADs). Even if the literature is still lacking in understanding the connection between immune system and ART procedures in ADs, recently, Simopoulou and colleagues [85] tried to investigate the role of several autoantibodies found in women affected by ADs undergoing IVF cycles as thyroiditis-related AA (TAA), anti-phospholipid antibodies (aPLs), anti-nuclear antibodies (ANA), AA affecting the reproductive system and AA related to celiac disease. Unexpectedly, aPLs, thyroiditis AA and anti-sperm antibodies (ASA) appeared to not be responsible for negative consequences in IVF cycles, even if their presence was associated with a higher miscarriage rate. By contrast, anti-endometrial antibodies (AEA) and ANA seem to be connected with lower clinical pregnancy rate, thus suggesting their putative adverse effect on pregnancies [85].

A deeper knowledge of the AA roles in pregnancy is thus needed which depicts a new frontier for pregnancy-related risk prevention and additional clinical studies are also required to better investigate this fascinating field of research in both natural and ART-driven pregnancies.

## 6. Natural Protection for Heterologous Implantation

### 6.1. Endocrine Function of the Corpus Luteum in Implantation, Placentation and the Risk of PE

The corpus luteum (CL) is a transitional organ representing the main source of steroid hormones, and vasoactive and angiogenic substances, which play a critical role in the early phase of pregnancy [86]. Embryo implantation derives from the combination of a competent blastocyst and proper uterine receptivity, and occurs in the middle-to-late luteal phase [87]. However, both progesterone (P) and estradiol (E2) secreted by CL play important roles in endometrial maturation, angiogenesis, vasodilation, and placentation steps [88]. Progesterone acts in the pre-decidualization process, in which the endometrium is renewed to a highly vascularized secretory pattern and produces secretions that enhance intrauterine environment to support embryo attachment and implantation. Moreover, E2 enhances the synthesis of nitric oxide and angiogenic factors, such as VEGF and placental growth factor (PlGF) [89]. In a recent study, Pereira and colleagues have shown how P metabolites may be correlated with the onset of pre-eclampsia [90]. Elevated levels of 20α-dihydroprogesterone, 3α5α20α-hexahydroprogesterone [HHP] and 3α20α-HHP were found in pregnant women at the 24–29 weeks of gestation that consequently developed the pre-eclampsia [91]. Concerning E2, several studies have shown that an excess or decrease in its concentration in the early stages of pregnancy may result in altered placentation and thus predispose the onset of PE [90].

### 6.2. The Angiogenic Factors

CL produces several metabolites with angiogenic or anti-angiogenic effects. In the early and late luteal phase, 16-ketoestradiol (16-ketoE2) and 4-hydroxyoestrone (4-OHE1) are, respectively, produced with a pro-angiogenic effect by stimulating VEGF production [92]. Since high levels of 16-ketoE2 were found in women with pre-eclampsia, it may act as a compensatory mechanism to placental insufficiency. 2-methoxyestradiol [2-ME2] is a metabolite of 2-hydroxyestradiol produced by CL with antiangiogenic activity. It acts at nuclear level where prevents transcription of VEGF-related genes [93,94]. Interestingly, low levels of 2-ME2 correlate with abnormal placentation. Paradoxically, one study has shown that 2-ME2 produces a low-oxygen environment that allows for the differentiation of the trophoblast into a phenotype with increased endovascular invasion [95]. Thus, low levels of 2-ME2 have been detected in the context of early onset of the pre-eclampsia. Moreover, plasma concentrations correlate with the severity of this disease; in fact, low plasma levels of 2-ME2 are related with high systolic blood pressure and proteinuria, which contribute to its intensive clinical management [96].

## 7. Conclusions

Infertility is both a medical and social condition that affects a large female population worldwide and is certainly a disease. It is related to several pathophysiological circumstances and its pathogenesis is often unclear, with relative indecision to establish the appropriate treatment choices. In the last three decades, many treatments have been introduced for this disease and the field of reproductive medicine is currently in rapid evolution for the employment of novel therapeutic approaches, primarily based on IVF techniques.

However, despite the fact that IVF treatments with own or donor gametes significantly increase the couple’s chances of becoming parents, they also carry several maternal and neonatal risks principally related in procedures using heterologous gametes. In fact, this practice seems to be associated with a higher rate of major risky events during pregnancy as recurrent miscarriage, infections and placental diseases such as gestational hypertension, pre-eclampsia and post-partum hemorrhage, as well as to several maternal–fetal complications due to the immune system interaction with the effects of the in vitro manipulation of gametes. Furthermore, both processing and culturing techniques of oocytes as both autologous or heterologous gametes, could result in altered gene methylation responsible of epigenetic derangements. Hence, in the era of precision medicine, to overcome these limitations, it would be desirable for the community of women undergoing IVF as well as for physicians, to test such a potential risk by exploring the integrity of gene methylation expression in both own or donated eggs.

In conclusion, the identification of several potential risks in ART procedures related to OD is essential in preventing possible fetal–maternal diseases and necessary for clinicians to address proper screening and surveillance during IVF related pregnancies.

## Figures and Tables

**Figure 1 ijms-24-13945-f001:**
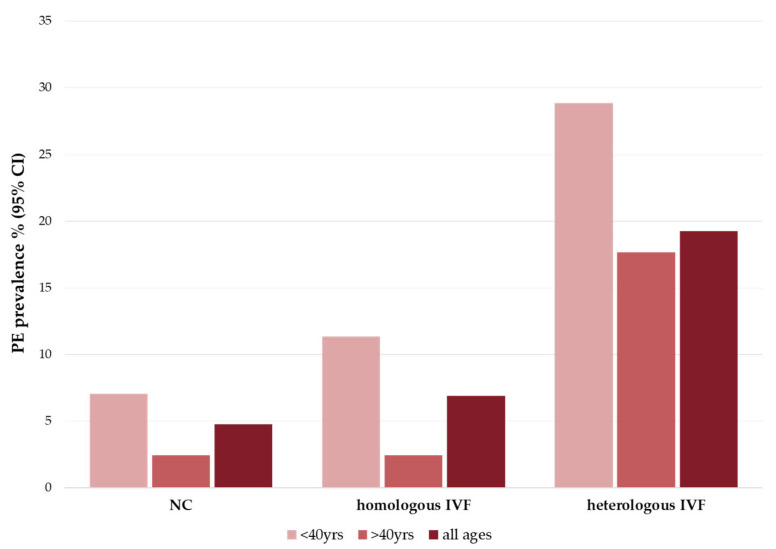
Pre-eclampsia prevalence in pregnancies. Histogram shows median value of pre-eclampsia (PE) prevalence percentage with 95% confidence interval (CI) in natural pregnancies (NC), homologous IVF pregnancies (with own gametes) and heterologous pregnancies (with donor gametes). PE outcome involves mild PE (blood pressure ≥140/90 mmHg in two or more occurrences, proteinuria ≥0.3 g/day over the 20th gestational weeks) and severe PE (blood pressure ≥160/110 mmHg in two or more occurrences, proteinuria ≥0.3 g/day or one of the following diseases: central nervous system dysfunction, hepatic abnormality, thrombocytopenia, renal abnormality or pulmonary edema). “All ages” column stands for the median value between <40 yrs and >40 yrs. Data were collected in a meta-analysis of 27 studies worldwide [33].

**Figure 2 ijms-24-13945-f002:**
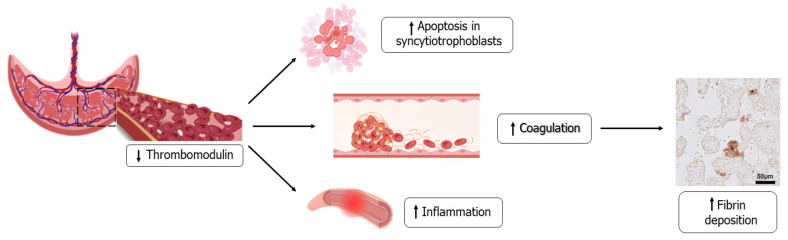
Abnormal placentation occurring in oocyte donation pregnancies. Placental thrombomodulin expression is lower in OD pregnancies than in natural ones, independently from the occurrence of pre-eclampsia and this reduction can be responsible for different complications such as apoptosis in syncytiotrophoblasts, increased inflammation and coagulation that then lead to increased fibrin deposition, as confirmed by immunohistochemical staining.

**Figure 3 ijms-24-13945-f003:**
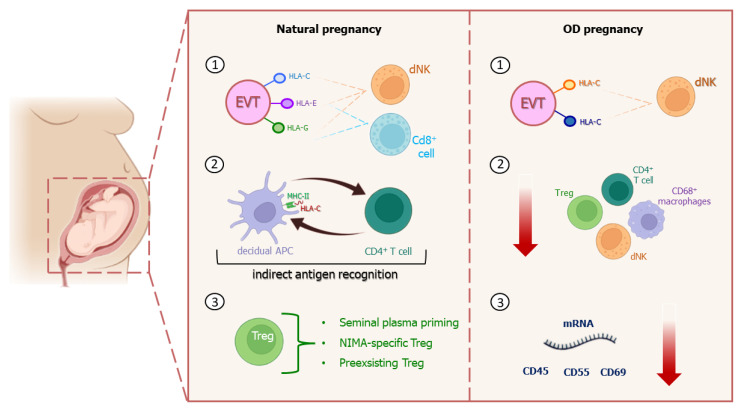
Immune system differences between natural and OD pregnancies. The main features of maternal immune response in OD pregnancies are the mother’s immune system overexposure to more non-self HLA-C antigens (paternal and donor’s ones) expressed on EVT surface (1); decreased expression of CD68^+^ macrophages, CD4^+^ T cells and Treg cells (2); and decrease in placental expression of several complement regulatory protein mRNA molecules (3).

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
