# Peer review of "Pregnancy by Oocyte Donation: Reviewing Fetal–Maternal Risks and Complications"

_ijms, 2023, doi:10.3390/ijms241813945_

Round 1
Reviewer 1 Report
The authors aim to review the fetal-maternal risks in pregnancy complications by oocyte donation. The chosen topic for the review is interesting; however, it raises valid concerns that warrant thorough exploration and careful consideration.
While the title and abstract of the article provide an intriguing overview of the topic, it's important to note that there appears to be a slight misalignment between the content discussed in the main body of the paper and the expectations set by the title and abstract.
This divergence raises questions about the focus and scope of the paper.
Also detailed description of the sources and methodology used to select the papers is not discussed.
Minor changes in English language is required.
Author Response
We are grateful to the Reviewer for helpful criticisms.
Comments and Suggestions for Authors:
The authors aim to review the fetal-maternal risks in pregnancy complications by oocyte donation. The chosen topic for the review is interesting; however, it raises valid concerns that warrant thorough exploration and careful consideration.
While the title and abstract of the article provide an intriguing overview of the topic, it's important to note that there appears to be a slight misalignment between the content discussed in the main body of the paper and the expectations set by the title and abstract.
Answer:
We are grateful to the Reviewer for his appreciation for our manuscript and for all comments. We thank also the Reviewer for this criticism and agree on the discrepancy between title, abstract and content of the main body.
We would like to emphasize that the manuscript was prepared in response to the Editor’s invitation for a contribution to a special issue on "Advances Techniques in Reproductive Medicine Research”. Therefore, we revisit the questioned topic of oocyte donation as practice of assisted reproductive technology (ART) focusing on the effective aspects of this technique in term of pregnancy rate, without missing the higher rate of major risky related events as recurrent miscarriage, infections and placental diseases, as well as several maternal-foetal complications due to the gametes manipulation and immune system interaction.
However, in the revised version of the manuscript we have now improved the abstract in order to better reflect the content and explain the scope of the manuscript in accordance with the Reviewer.
In consideration that ART is constantly advancing in terms of practical application as well as in knowledge, principally for many innovative studies and clinical trials carried through the years, we have considered in our manuscript the scientific documents published in the last 15 years, in order to provide evidence of the medical progress in the field of reproductive health.
Finally, we have now corrected the grammar errors around manuscript as requested.
Reviewer 2 Report
The manuscript entiled " Pregnancy by oocyte donation: reviewing fetal-maternal risks and complications" was read with great interest. The authors have done what appears to be an in depth review of the subject of the risks to the fetus and mother in oocyte donation.
There are a few suggestions that may help clarify points in the article.
Introduction
Page 1 line 38- Suggest context versus content.
3. Major Risky events
page 3 line 134- Suggest Thanks versus Thank,
Page 3 lines 143-46- This statement is not timely. It appears too strong in line with current data. The article is >16 years old and IVF is much different in regards to stimulation, lab techniques and media, and Fresh vs. frozen transfers. In addition, there have been several RCT's that have not supported the findings in this meta-analysis. Suggest delete. If there is more recent data, it should be used.
Page 4 line 151-154- Reference is older and would suggest softening statement to ...needs to be further evaluated as an approach to prevent...
Figure 1 is clear and makes point.
3,2 IVF and fetal-maternal risk
Page 5 line 221-224- This reference is 25 years old . IVF has changed as noted previously. Suggest deleting this sentence or finding a recent article.
Risk of abnormal placentation
Interesting discussion with many of the proposed ideas.
Page 8 line 353- Suggest delete the reference #81 as it is old and was only an abstract.
Also the soft statement at end of section is appropriate.
Natural protection...
Page 9 line 399- Should this reference be 92 versus 90? If yes, is reference 90 used elsewhere?
None
Author Response
We thank are the Reviewer for comments and suggestions.
- Page 1 line 38- Suggest context versus content.
Answer: We have now replaced content with context (page 1, line 38).
- Page 3 Line 134 Suggest Thanks versus Thank
Answer: We have changed Thank with Thanks (page 3, line 134).
- Page 3 lines 143-46- This statement is not timely. It appears too strong in line with current data. The article is >16 years old and IVF is much different in regards to stimulation, lab techniques and media, and Fresh vs. frozen transfers. In addition, there have been several RCT's that have not supported the findings in this meta-analysis. Suggest delete. If there is more recent data, it should be used.
Answer: We have now replaced reference 26 with a more recent one (Page 3, line 143). Hence, the text has been modified as following: “Despite inconsistent conclusions derived from meta-analyses [24, 25] exploiting IVIg in RM patients, it seems to represent a chance for couples that failed to achieve a succesful pregnancy after IVF and/or in "unexplained" infertility cases, often related to immunological problems, as reported by Virro [26]. However, further studies will be necessary before considering IVIg clinical employment“ (Page 3, lines 140-144).
- Page 4 line 151-154- Reference is older and would suggest softening statement to ...needs to be further evaluated as an approach to prevent...
Answer: As suggested we mitigate our sentence from "prednisone seems to be a promising approach to prevent pregnancy loss" to "prednisone needs to be further evaluated as an approach to prevent recurrent pregnancy loss" (page 4, lines 151,152).
- Page 5 line 221-224- This reference is 25 years old. IVF has changed as noted previously. Suggest deleting this sentence or finding a recent article.
Answer: As suggested we modified reference 48 with a recent one (Page 5, line 220). As a consequence, we also modified the text as following: ”Anyway, according with previous studies, a recent work of Zhu and colleagues [48] show that, considering IVF and ICSI as different ART subgroups, there are no risks differences for birth defects between children born with one or the other technique” (Page 5, lines 219-222).
- Page 8 line 353- Suggest delete the reference #81 as it is old and was only an abstract.
Answer: We have deleted the reference 81 as suggested. The following reference numbers has been changed to maintain the progression (Page 8, line 351).
- Page 9 line 399- Should this reference be 92 versus 90? If yes, is reference 90 used elsewhere?
Answer: We thank the Reviewer for this criticism and agree on the discrepancy between reference 90 and the relative text, which is instead referred to reference 92. Anyway, since we delete reference 81 as written before, and we also must maintain references progression, the old reference 90 is now 89 (moved to page 9 line 396) and the old references 92 are now 90 (page 9 lines 397 and 402, respectively).